# Lipid-Laden Microglia: Characterization and Roles in Diseases

**DOI:** 10.3390/cells14161281

**Published:** 2025-08-19

**Authors:** Jiani Xing, Takese McKenzie, Jian Hu

**Affiliations:** 1Department of Cancer Biology, The University of Texas MD Anderson Cancer Center, Houston, TX 77054, USA; jxing1@mdanderson.org (J.X.); ttmckenzie@mdanderson.org (T.M.); 2Cancer Biology Graduate Program, The University of Texas MD Anderson Cancer Center UTHealth Houston Graduate School of Biomedical Sciences, Houston, TX 77030, USA; 3Neuroscience Graduate Program, The University of Texas MD Anderson Cancer Center UTHealth Houston Graduate School of Biomedical Sciences, Houston, TX 77030, USA; 4Cancer Neuroscience Program, The University of Texas MD Anderson Cancer Center, Houston, TX 77030, USA

**Keywords:** lipid droplets, microglia, neurodegeneration, lipid metabolism, lipid droplet analytical tools

## Abstract

Microglia are resident phagocytes of the central nervous system that play an essential role in brain development and homeostasis. When the intracellular lipid content exceeds the metabolic capacity of microglia, lipid droplets accumulate, giving rise to a distinct population termed lipid-laden microglia (LLMs). LLMs have been implicated in various neuroinflammatory and neurodegenerative diseases, functioning as both regulators/indicators of inflammation and potential therapeutic targets. This review summarizes the current research on LLMs, focusing on disease-specific regulators and functions, protective roles, interactions with neighboring cells, and advances in diagnostic and analytical tools. We also discuss the blurred distinction between LLMs and macrophages, inconsistent terminology, and major knowledge gaps across different disease contexts. Deciphering the composition, formation, and dynamics of lipid droplets in microglia is critical for uncovering how microglial states shift under diverse pathological stimuli. A clearer view of these mechanisms may reveal novel roles of LLMs and open new avenues for therapeutic intervention.

## 1. Introduction

From unicellular organisms to humans, lipids are essential biomolecules that serve as structural components, energy reservoirs, and signaling mediators in cells [1]. For many years, lipid droplets (LDs) were believed to be inert fat particles used to store energy. However, accumulating evidence has redefined LDs as dynamic, endoplasmic reticulum (ER)-derived organelles involved in lipid metabolism, stress response, and signal transduction [2]. LDs contribute to lipid and energy homeostasis and participate in modulating ER stress, oxidative stress, and various signaling pathways [3,4]. Their biogenesis and structural features appear to be conserved across species and cell types [5]. Structurally, LDs contain a neutral lipid core—mainly triacylglycerols, diacylglycerols, monoacylglycerols, and cholesteryl esters (CEs)—encased in a phospholipid monolayer embedded with proteins such as perilipin (PLIN) and seipin [5,6].

The observation of glial lipid accumulation dates to 1907, when Alois Alzheimer described such changes in the brains of dementia patients [7]. Decades later, in the 1970s, lipid-laden macrophages—termed “foamy macrophages”—were identified as markers of immune dysfunction in atherosclerotic lesions [8]. Since then, the view of LD accumulation as a mere byproduct has been replaced by recognizing its role as a disease marker and active player in central nervous system (CNS) dysfunction. Notably, LD accumulation in microglia has been implicated in various neuroinflammatory and neurodegenerative diseases. Microglia are phagocytes that play an essential role in brain development and homeostasis. The accumulation of LDs in microglia is disease-specific and can be triggered by various pathological stimuli, such as lipopolysaccharide, long-chain fatty acids, brain injury, aging, amyloid-beta (Aβ), and even tauopathy neurons [9]. These triggers activate distinct signaling pathways, contributing to diverse microglial phenotypes and functional states. This review highlights recent progress in the characterization, regulation, and function of microglia with lipid droplets across diseases—including neurodegeneration, cancer, and metabolic disorders—and discusses their emerging diagnostic value and therapeutic potential.

## 2. Lipid-Laden Microglia in the Context of Microglial Heterogeneity

Microglia are the resident phagocytes of the central nervous system that promote homeostasis. Unlike monocyte-derived macrophages, microglia originate from the embryonic yolk sac and migrate into the neural tube, where they undergo maturation [10]. Initially, microglia were categorized into M1 and M2 states. The M1 state was proinflammatory and neurotoxic, while the M2 phase was anti-inflammatory and neuroprotective [11]. However, the field has now acknowledged that microglia cannot be categorized into two distinct states because these cells are highly dynamic, and their function is highly context-dependent [12]. Microglia are known to acquire diverse functional states during development and disease [12]. Homeostatic microglia survey the brain environment, phagocytose debris, and secrete regulatory cytokines. These microglia are characterized by markers such as P2RY12 and TMEM119 [12]. Microglial states are dynamic as microglia transition from a homeostatic state to different functional states in disease, aging, and injury, giving rise to microglial heterogeneity [12]. This diversity ranges from disease-associated microglia (DAM) found in Alzheimer’s Disease to white matter-associated microglia (WAM) in demyelination and aging. These states are characterized by the expression of specific gene signatures; for example, DAMs express markers such as Clec7a and *APOE*. Together, this shows that the role and state of microglia are highly heterogeneous and context-dependent.

Over the years, studies have identified the distinct accumulation of lipid droplets in a subpopulation of phagocytes across various disease models. In studies on demyelination, these cells are referred to as “foamy phagocytes,” which describes cells with a bubbly, lipid-rich appearance near demyelinated plaques [13]. Originally identified in atherosclerotic lesions, foamy phagocytes indicate that cells have engulfed excessive lipids—particularly cholesterol and cholesteryl esters—leading to the formation of LDs. [8,14]. These LD-enriched cells are considered a hallmark of atherosclerosis [15]. While phagocytes such as macrophages dominate the foamy cell population, other cell types (e.g., microglia, smooth muscle cells, and endothelial cells) can also become foamy [14]. Microglia are the resident phagocytes of the brain, and recently, studies have characterized the presence of LD accumulating in microglia in the brain. More recently, the term “lipid droplet-accumulating microglia” (LDAM) was introduced to describe a subtype of microglia in aged and Alzheimer’s Disease (AD) brains that accumulate triglyceride-rich LDs and display impaired phagocytosis and heightened oxidative stress [6]. In other studies, terms like “lipid droplet-rich microglia (LDRM)” are used interchangeably with ‘foamy microglia’ and ‘lipid-laden microglia’, adding to the terminological ambiguity [16,17,18,19]. While extensive efforts have been made to characterize microglial functional states, other than the accumulation of lipids, very little is known about the gene expression profile of microglia with extensive LDs. Lipid metabolism genes regulate the formation of LDs, and many genes involved in lipid metabolism are dysregulated across multiple microglial phenotypes, including DAMs [12]. Outside of AD and aging, whether lipid droplet-rich microglia represent a unique microglial subtype or a continuum of a pre-existing microglial subtype has yet to be fully elucidated. For this review, we have adopted the umbrella term “lipid-laden microglia” (LLM) to collectively describe microglia with intracellular LD accumulation, regardless of disease trigger or lipid species. While this generalization facilitates discussion, it underscores the urgent need for future studies to classify LLM subtypes based on their molecular triggers, functional phenotypes, gene expression, and lipidomic profiles.

## 3. Molecular and Analytical Tools for Detecting Lipid Droplets in Microglia

### 3.1. Traditional Detection Methods (Staining and Labeling)

The traditional method of detecting LDs in cells relies on staining intracellular neutral lipids using lipophilic dyes like Oil Red O (ORO), Sudan Black B, BODIPY, and Nile Red. ORO and Sudan Black B staining, in combination with conventional wide-field trans-illumination microscopy, have been widely used to visualize LD morphology [20,21]. The traditional ORO staining protocol was later refined by reducing the staining time to 1 min and the fixation time to 10 min, achieving a clearer visualization of LDs in over 90% of macrophages [21]. The use of 60% isopropanol during both infiltration and destaining enhanced lipid penetration and minimized background staining, improving imaging quality [21]. Oxidized low-density lipoprotein (oxLDL) is internalized by microglia and macrophages via scavenger receptors and accumulates in phagolysosomes, promoting the transformation of these cells into foamy phenotypes under ischemic conditions [22]. DiI, a lipophilic and non-toxic fluorescent dye, can be conjugated to oxLDL (DiI-oxLDL) and detected by confocal microscopy or flow cytometry. This approach enables quantitative analysis of oxLDL uptake and subsequent LD accumulation, offering a functional tool for studying the formation of lipid-laden cells [21].

However, ORO and Sudan Black B staining are restricted to fixed samples and highly sensitive to preparation conditions, making them time-consuming and less reproducible [20]. In contrast, BODIPY and Nile Red are lipophilic fluorescent dyes with high specificity and photostability, which are suitable for both live and fixed cell imaging. While the autofluorescence of lipofuscin (480–695 nm) partially overlaps with the emission of BODIPY 493/503 (∼505 nm), it is demonstrated that lipofuscin requires longer exposure times (5–10 s) to become visible, whereas BODIPY imaging requires only ∼750 ms of exposure [23]. This difference minimizes interference and validates BODIPY as a reliable LD marker in microglia [23]. Nile Red displays polar-sensitive spectral shifts, changing from yellow (580 nm) for neutral lipids to red (628 nm) for polar lipids. These spectral properties can be both a challenge and an opportunity: while the color change complicates quantitative imaging, it can also indicate a change in the composition of the LDs [20].

Another widely used approach to visualize LDs across a broad size range is electron microscopy, including transmission and scanning electron microscopy (TEM and SEM). These techniques provide a subcellular resolution ranging down to <10 nm, allowing for the precise visualization of LD localization and their spatial relationships with other intracellular structures [20]. In TEM, a beam of high-energy electrons (80–300 keV) is transmitted through ultrathin tissue sections. The small electron wavelengths enable resolution at the angstrom scale. As primary electrons interact with the sample, they reveal the internal ultrastructure in detail. Due to its superior spatial resolution, TEM is a valuable tool for studying microglial LDs. It has been used to detect LDs in various pathological conditions, such as Alzheimer’s disease (AD), multiple sclerosis (MS), and spinal cord injury (SCI). Under pathological stress, microglia can acquire a distinct ultrastructural phenotype known as dark microglia, characterized by electron-dense cytoplasm and nuclear condensation [24]. In TEM images, LDs in these cells appear as bright, spherical structures enclosed by a single membrane. It has been shown that single dark microglia can contain up to 100 LDs [25]. Given the ability of TEM to resolve the microglial ultrastructure, it is an indispensable tool for characterizing LLMs.

Furthermore, confocal fluorescence microscopy combined with immunohistochemistry is widely used to detect intracellular LDs and their associated proteins, such as seipin, PEX30, and members of the PLIN family [26]. While these traditional techniques are straightforward, cost-effective, and widely accessible, they generally require the chemical fixation or staining of cells, limiting their ability to monitor dynamic LD behavior under physiological conditions. Moreover, excess fluorescent dye can result in nonspecific labeling and potentially interfere with cellular lipid metabolism [27].

### 3.2. Label-Free Methods

Given the limitations of label-dependent methods—including fixation requirements, staining artifacts, photobleaching, and spectral overlap—label-free imaging techniques are gaining attention for studying LDs in a more physiologically relevant manner. For example, synchrotron-based micro-Fourier transform infrared (microFTIR) spectroscopy can be employed to analyze foamy cells derived from RAW264.7 macrophages [28]. This technique enabled the detection of lipid content and efflux in response to experimental treatments without the need for fluorescent or chemical labeling [28]. At the single-cell level, coherent anti-Stokes Raman scattering (CARS) microscopy and stimulated Raman scattering (SRS) microscopy offer high-throughput, label-free, non-invasive, and chemically specific imaging by detecting the intrinsic vibrational frequencies of molecular bonds, particularly C–H stretches in lipids. Notably, studies show that under comparable imaging conditions, CARS can provide superior spatial resolution relative to SRS [27,29], while SRS offers improved signal linearity and quantification. Another emerging label-free technique is holotomography, also known as optical diffraction tomography (ODT), which reconstructs three-dimensional (3D) images based on refractive index differences between LDs and the cytoplasm [30]. ODT allows for real-time, live-cell 3D imaging of LD volume, number, distribution, and total lipid content. 

Recently, the emergence of artificial intelligence has opened new avenues for exploring LDs using label-free imaging approaches. Coupling ODT with a deep learning-based virtual labeling framework allows for the simultaneous prediction of multiple fluorescent markers from a single refractive index scan, achieving the real-time, label-free 4D tracking of LDs in live cells [31]. Overall, the label-free methods avoid many drawbacks of traditional staining techniques, such as phototoxicity, photobleaching, and nonspecific labeling [32]. They offer a powerful and precise platform for investigating lipid metabolism, dynamics, and storage in single live cells with minimal perturbation. Both the label-dependent and label-free tools discussed in this section are summarized in Table 1, along with their respective advantages and disadvantages.

## 4. Lipid-Laden Microglia Formation in Aging

Aging is one of the strongest risk factors for many neurodegenerative diseases [33]. It causes the dysfunction of cells and is characterized by various hallmarks, including telomere attrition, genomic instability, epigenetic alterations, altered intercellular communication, stem cell exhaustion, cellular senescence, mitochondrial dysfunction, dysregulated nutrient sensing, and loss of proteostasis [33]. Microglia, a key cell type important for regulating brain homeostasis, are also dysregulated during aging [34]. Aging also induces the dysregulation of microglial function and lipid metabolism; therefore, it is reasonable to infer potential alterations in LLMs during aging [34]. Earlier, a significant increase in lipid-laden cells in the aged brain was found using ORO staining [35]. Recently, a pioneering study was published, reporting a striking buildup of LDs in microglia in aged mouse and human brains [6], termed Lipid Droplet-Accumulating Microglia (LDAM). LDAMs are characterized as having defective phagocytosis, increased proinflammatory cytokine secretion, and elevated levels of reactive oxygen species. These characteristics implicate LDAMs as possible instigators of dysfunction during aging.

Studies have sought to identify specific mediators of increased microglial reactivity with aging and focused on sialic acid-binding immunoglobulin-like lectin-11 (SIGLEC-11), a human microglial surface receptor that dampens microglial inflammatory pathways [36]. Expressing human SIGLEC-11 in mice and examining these mice at 6 and 24 months old, they found reduced LLM numbers, neuronal loss, and the expression of proinflammatory genes, suggesting that SIGLEC-11 attenuated neuroinflammation and the formation of LLM with age.

Lifestyle factors, especially diet, are modifiable aspects that can affect aging. A high-fat diet promotes inflammation and perturbs lipid metabolism, and its consumption is increased in older adults [37]. To assess the impact of diet on brain function during aging, aged mice were fed with a high-fat diet [38]. A significant increase in neutral lipid content, assessed by BODIPY and microglial lipid load, was observed in these mice. Microglia in the hippocampus were also more activated, as seen by an increase in CD68 immunoreactivity. Furthermore, another study shows that PPAR agonists were able to reduce microglial LDs and partially restored microglial function [39]. Together, these studies highlight that, during aging, the burden of LLMs increases, and these microglia tend to be more proinflammatory, thus possibly driving dysfunction.

## 5. Lipid-Laden Microglia in CNS Neurodegeneration

Neurodegeneration refers to the chronic progressive decline in the central nervous system [40]. Microglia, which constitute about 10% of the brain’s cells and serve as its resident immune cells, are key mediators of neurodegeneration [41]. In this section, we summarize the current literature detailing the role of LLM in neurodegenerative diseases.

### 5.1. Alzheimer’s Disease

AD is one of the most common forms of dementia and a leading cause of death and disability in the elderly worldwide [42]. The key hallmarks of AD include the aggregation of Aβ plaques, hyperphosphorylation of tau, neuronal loss, and neuroinflammation [43]. Some of the strongest genetic risk factors for AD include mutations in *APOE* and *TREM2*, which are microglia-enriched genes that regulate lipid metabolism [43]. Given the importance of lipid metabolism in the formation of LLMs, it is expected that LLMs are present in AD.

LDAMs, defined by the expression of the LD-associated enzyme acyl-CoA synthetase long-chain (*ACSL*), were recently found in the AD brain [44]. *ACSL1+* microglia were more abundant in AD patients with the *APOE4/4* genotype, and carrying the E4 mutation in the *APOE* gene has been identified as one of the most significant risk factors for the development of late-onset AD [45]. The abundance of LD-laden *ACSL1+* microglia in *APOE4/4* brains suggests that dysfunction in *APOE* lipid metabolism could drive the formation of LDAMs in AD. Additionally, *APOE4/4* iPSC-derived microglia that contain LDs induce tau hyperphosphorylation and neuronal apoptosis, highlighting the detrimental nature of these microglia.

*TREM2* is a myeloid cell receptor that is significantly upregulated in microglia associated with amyloid plaques [46,47]. *TREM2* has also been defined as a key activator of the disease-associated microglia found around plaques [47]. The presence of the *TREM2-R47H* variant is also associated with an increased risk for the development of AD [48]. Claes et al. addressed the effect of the *TREM2-R47H* mutation on LD formation in AD by transplanting *TREM2-R47H* iPSC-derived microglia into a chimeric AD mouse model. In vitro, these microglia displayed an accumulation of LDs; however, in vivo, the opposite was found. This suggests that LD accumulation occurs secondary to *TREM2* mutation.

In addition to specific genetic risk factors, LDAMs can be mediated by changes in the brain environment due to AD pathology. Prakash et al. [49] demonstrated that Aβ exposure induces the formation of LDs in microglia in a proximity-driven manner in AD. Similarly to previous reports, these microglia exhibited defective Aβ phagocytosis. The study suggests that the accumulation of LDs is mediated by changes in the microglial lipid composition through decreasing free fatty acids and increasing triacylglycerols. Inhibiting diacylglycerol O-acyltransferase 2, an enzyme that converts free fatty acids to triacylglycerols, promoted microglial Aβ phagocytosis and reduced neuronal damage.

Additionally, the inhibition of fat storage-inducing transmembrane protein (FIT2) reduced LD formation and improved the microglial phagocytosis of Aβ plaques [50]. In assessing the formation of LDs in AD microglia, Sha et al. [51] utilized the 3xTg AD mouse model, which has both amyloid and tau pathology. The LDs in AD microglia were regulated through the TRPV1–PKM2–SREBP1 axis. PMK2 and SREBP1, which are particularly enriched in microglia, were upregulated in the 3xTg mice, accompanied by increased LD formation in these mice. In this study, it was established that treating AD mice with capsaicin increased microglial phagocytic function through the inhibition of PKM2 dimerization and reduction in SREBP1 activation, which in turn increased TRPV1 activation. These findings suggest that modulating microglial lipid droplet formation is a potential therapeutic for AD.

Complement C3a receptor (C3aR), which is predominantly expressed in microglia, was also found to be a regulator of lipid accumulation in AD mice [52]. In the amyloid precursor protein (APP) AD mouse model, microglia shifted to a high-C3aR-expressing subpopulation, and the deletion of C3aR attenuated LD accumulation in microglia. Using an in vitro cell culture system with BV2 immortalized microglial cells, Li et al. [53] showed that Aβ treatment directly induced the formation of LDs. This profile was strongly correlated with the increased expression of the lipid metabolism gene *ANGPTL4*. Additionally, others have uncovered a unique regulation of neuronal activity by LD-accumulating microglia [54]. Human-derived microglia expressing the *APOE4* allele were found to have an accumulation of LDs. These microglia had a weak response to neuronal activity. Exposing neurons to conditioned media from *APOE4* LLMs also led to a significant decrease in neuronal calcium transients.

These results support the AD environment and lipid metabolism genes as key drivers of LD accumulation in microglia. Given the role of LD-accumulating microglia in AD, researchers have tried to therapeutically target them. Comerota et al. [55] utilized oleoylethanolamide, a health span-promoting endogenous lipid amide, as a therapeutic agent in chronically inflamed 5× FAD mice [56]. Oleoylethanolamide was able to specifically reduce LD formation in the microglia through a PPARα-dependent mechanism. A study by Wu et al. also demonstrated how the modulation of microglial lipid metabolism through fasting could alter LD formation in AD [50]. APP/PS1 mice that underwent intermittent fasting demonstrated decreased accumulation of LDs in the microglia. These microglia also showed increased Aβ phagocytosis and a more amoeboid shape. Intermittent fasting also improved cognitive function, as measured by the Barnes maze and Y maze.

Overall, an abundance of studies have shown that LD accumulation in microglia is a predominant feature of AD. The accumulation of LDs in AD causes these microglia to become defective, exhibiting reduced phagocytosis and increased proinflammatory signals. The formation of LLMs in AD is influenced by genetic risk factors such as *APOE4* and *TREM2-R47H* mutations. Additionally, perturbations in lipid metabolism genes such as *ACSL*, *DGAT2*, and *FIT2* also significantly contribute to the formation of LLMs.

### 5.2. Tauopathies

Tauopathies refer to a diverse group of neurodegenerative diseases that are characterized by the abnormal accumulation of tau in the brain. Tau is a microtubule-associated protein that functions by binding to tubulin, promoting its polymerization and stabilization to form microtubules [57]. Tauopathies are classified into two subsets: primary and secondary [57]. Primary tauopathies refer to diseases where tau deposition is the predominant feature. Secondary tauopathies refer to diseases where another upstream factor drives the deposition of tau. Tauopathies include AD, Pick’s disease, and progressive supranuclear palsy. The predominant clinical features of tauopathies are progressive aphasia, cognitive deficits, and movement disorders.

The accumulation of tau leads to chronic neuroinflammation in the brain. Li et al. [58] utilized Raman scattering (SRS) microscopy to visualize the brains of mice with tauopathy and observed the striking accumulation of LDs in phagocytes, marked by ionized calcium-binding adaptor molecule 1 (Iba1) and CD68. These LDs originated from neurons, which were subsequently transferred to phagocytes in the mouse hippocampus. These phagocytes with LDs had features like LDAMs found in AD. The study found that LD accumulation in microglia in the presence of tau results in defective phagocytosis and the secretion of proinflammatory cytokines. While many studies have used Iba1 and CD68 to mark microglial cells, these markers are also expressed in monocyte-derived macrophages [59], which have functional roles in tauopathies [60].

On assessing publicly available RNA sequencing data on tauopathy human brains, Li et al. found a striking dysregulation of genes involved in LD formation, including PLIN2, GPAT1, and ABCA1. Neuronal AMP-activated protein kinase (AMPK), a master regulator of energy homeostasis, was also significantly downregulated, and its depletion in iPSC neurons co-cultured with BV2 microglia promoted the accumulation of LD in microglia. Tau in the presence of microglia was also found to directly increase the abundance of phospholipids and sphingolipids, thus facilitating the accumulation of LDs [61]. Additionally, a study published by Xu et al. [62] further delineated the role of LD-accumulating microglia in tauopathy. They characterized the role of the autophagy gene Atg7 in microglia/ macrophages using a Cx3CR1 promoter-driven conditional knockout mouse model and cultured BV2 immortalized microglial cell line. The disruption of autophagy caused by Atg7 depletion shifted microglia to a proinflammatory state with heightened LD accumulation. In addition, Atg7-depleted microglia also promoted the spread of tau in neurons [62]. Together, these findings indicate that tau can directly induce LD formation in microglia, which is detrimental to the microglia. They also highlight the dynamics of microglia, whereby LD formation can be induced by adjacent tau-laden neurons and dysfunction in intrinsic microglia autophagy.

### 5.3. Demyelinating Diseases 

Demyelination refers to the pathological loss of myelin, a lipid-rich structure in the nervous system [63]. Myelin comprises about 70% lipids, including cholesterol, phospholipid, and glycolipid [63]. The phagocytosis of lipid-rich myelin by phagocytes leads to the acquisition of a foamy phenotype. This foamy phenotype is characterized by the presence of LDs rich in cholesteryl esters surrounded by a phospholipid monolayer. Most of the published literature has focused on LD accumulation in Multiple Sclerosis (MS) and Krabbe’s disease; therefore, we summarize the findings on these demyelinating diseases below.

MS is the most common primary demyelinating neurodegenerative disease, affecting over 2 million people worldwide. It is characterized by focal demyelinating lesions in the brain and spinal cord that lead to motor and neurological impairments [64]. The lesions are characterized by the accumulation of LD-laden foamy phagocytes, infiltrating T-cells, neuronal loss, and damaged myelin. Studies have also shown that microglia are not the only phagocytes present in the lesions, as monocyte-derived macrophages also infiltrate from the periphery. However, MS involves demyelination of the CNS; therefore, microglia, the resident phagocytes of the CNS, are thought to have a predominant role.

In a study characterizing early MS plaques, 60% of phagocytes marked by EBM11 (CD68) expressed nucleoside diphosphatase activity, a microglial marker [65]. This indicates that microglia are the main phagocytes in early MS and thus play an important role in disease progression. While this study has delineated the origin of the main phagocytes in MS, very few studies have made a clear distinction between the role of yolk sac-derived microglia and monocyte-derived macrophages. This is highlighted in a study where the term ‘microglia’ was used to define the entire population of phagocytes in the MS demyelinating lesion [66]. Therefore, there is a need for future research to clearly distinguish the contribution of lipid droplets in microglia and macrophages to MS pathology.

Given the prevalence of foamy phagocytes in MS pathology, researchers have characterized their regulators. One study found that Chitinase 1 (CHIT1) is mostly expressed by lipid-laden microglia/macrophages in active postmortem MS lesions [67]. CHIT1 is an enzyme that facilitates the degradation of chitin-containing pathogens. CHIT1 expression was correlated with the transition of microglia/macrophages to a more activated state and associated with foam cell differentiation. Additionally, CHIT1 cerebrospinal fluid (CSF) levels correlated positively with later disability. Together, these data suggest that CHIT1 produced by lipid-laden microglia/macrophages is a potential biomarker for disability progression in MS.

In addition, CSF levels of neurofilament light chain (NfL) were also strongly correlated with the proportion of MS patient lesions containing foamy microglia/macrophages [68]. The active lesions with foamy microglia/macrophages also had a higher proportion of axonal damage, suggesting increased neurodegeneration. Conversely, CSF NfL levels were negatively correlated with the proportions of inactive and remyelinating lesions. Together, this study suggested that the presence of foamy microglia/macrophages may be a driver of neurodegeneration, and CSF NfL could be a biomarker of foamy microglia/macrophage-driven lesion activity and disease progression in MS.

The uptake of myelin drives the formation of foamy microglia, and scavenger receptors bind to and facilitate the phagocytosis of myelin. Hendrickx et al. [69] characterized changes in the expression of the scavenger receptors CXCL16, SR-AI/II, FcyFRIII, and LRP-1 in the chronic active lesions of microglia of MS patients [69]. These microglia were defined by the expression of HLA-DR, Iba1, and CD68 [58]. The scavenger receptors were specifically upregulated in the foamy microglia at the rims of chronic active MS lesions, suggesting that they mediate early phagocytosis and the formation of foamy microglia during demyelination. The dynamics of these foamy microglia are also regulated by several players during demyelination. Using the lysolecithin mouse model of demyelination, Ma et al. [70] showed that miR-223 regulated the degradation of LDs through lipophagy in microglia. Microglia were characterized using Iba1 staining and transmission electron microscopy imaging. Specifically, miR-223 enhanced autophagy, thereby reducing LD accumulation in microglia by inhibiting cathepsin B. Together, this inhibited the formation of LLM during demyelination.

In a case of demyelination induced by ischemia, Low-Density Lipoprotein (LDL) promoted lipid accumulation in microglia [71]. Microglia activated by LDL were shown to be detrimental by promoting the breakdown of the blood–brain barrier and ischemic demyelination. PLIN2, a surface marker of LDs highly expressed in lipid-laden microglia/macrophages in many demyelination diseases, helps LDs escape from degradation, which impairs the remyelination process. PLIN2 deficiency reduced inflammation signals and accelerated remyelination, providing new insight into the treatment of demyelinating disease with LDs accumulation [72]. These studies together highlight the strong correlation between key proteins expressed during the formation of LLMs and disease progression in MS.

Disrupted lipid metabolism directly affects the formation of LLMs; therefore, researchers have targeted this dysmetabolism to alter microglial profiles. A demyelinating disease characterized by LLMs and macrophages is Krabbe’s disease. This disease is caused by loss-of-function mutations in galactosylceramidase, which results in neuroinflammation, demyelination, and neurodegeneration [73]. Aisenberg and colleagues showed the paramount role of microglia in Krabbe’s disease [74] by demonstrating that the replacement of galactosylceramidase-mutated microglia with healthy microglia was able to attenuate disease progression in mice. This highlights the role that LLMs play in promoting neurodegeneration.

On the other hand, others have demonstrated that promoting the formation of LLMs can help prevent demyelination [55]. They treated a rodent model of inflammatory injury with UCM1341, a compound that inhibits fatty acid amide hydrolase and activates melatonin receptors. These targets have been previously shown to have potential for treating neuroinflammation-driven neuropathological states. UCM1341 stimulated the formation of lipid-laden microglia/macrophages and attenuated demyelination. The protective role of LDs was also demonstrated by the identification of *TREM2* as a key mediator of LD formation expressed in microglia and macrophages. The loss of *TREM2* led to a disruption in LD formation, increased ER stress, and reduced remyelination [75]. This study indicated that microglia/macrophages clear the excess lipid from myelin breakdown by generating LDs in a *TREM2*-dependent way, which is a necessary process.

Overall, lipid-laden microglia/macrophages are a key defining feature of demyelinating lesions. They have cholesterol-rich LDs from the phagocytosis of myelin and are initially beneficial for disease recovery through the clearance of damaged myelin. However, the chronic presence of foamy microglia/macrophages is linked to increased demyelination. Regulators of foamy phagocyte formation include enzymes (CHIT1), scavenger receptors (CXCL16), and microRNAs (miR-223). The expression of some of these markers in the CSF has been used as a biomarker for disease progression. A key deficit in the field of demyelination is the lack of a functional distinction between lipid-laden microglia and macrophages. Common markers that are widely used include Cx3CR1 and Iba1, which are unable to differentiate between microglia and macrophages. Future studies could address this by incorporating microglia-specific deletion of lipid metabolic genes and microglia-specific markers such as P2Y purinoceptor 12 (P2RY12).

## 6. Lipid-Laden Microglia in CNS Injury

Central Nervous System (CNS) injury refers to an insult to the brain or spinal cord that results in neuronal dysfunction and disability. As the resident immune cells of the brain, microglia represent one of the first responders to CNS injury. Here, we summarize the role of LLM in spinal cord injury and traumatic brain injury, the two most studied CNS injuries in the literature. 

### 6.1. Spinal Cord Injury

Spinal Cord Injury (SCI) affects millions worldwide and has devastating consequences, including impaired motor and nerve function, chronic pain, and severely reduced quality of life [76]. The pathology of SCI includes necrotic cell death, the accumulation of myelin debris, the activation of microglia, and the infiltration of macrophages [77,78]. The spinal cord is a part of the CNS composed of long, parallel neuronal axons wrapped with myelin [77]. SCI results in damage to myelin, which is subsequently cleared by resident microglia and macrophages to attenuate neuroinflammation. The phagocytosis of cholesterol-rich myelin results in the formation of LDs and the induction of the foamy microglial/macrophage state. The accumulation of LDs in microglia is a prevalent feature of SCI [78]. The presence of LDs in microglia during SCI has been characterized utilizing CARS microscopy, a technique that is superior for visualizing myelin and lipid bodies in biological tissue [77]. Using this technique in a rat model of SCI, LLMs were successfully detected in inflammatory SCI lesions. 

Recently, work has been carried out to characterize the regulators of microglial LDs in SCI. Ou et al. [78] utilized the hemicontusion SCI model established by using an impactor tip to form a contusion at the C5 lamina in the mouse spinal cord. Using TEM coupled with TMEM119 as the marker to specifically label microglia, they identified that LDs were predominantly accumulated in microglia in the epicenter of the injury. RNA sequencing of the injured spinal cord revealed a significant upregulation in pathways such as microglial cell activation, phagocytosis, and lipid homeostasis. ATP-binding cassette transporter A1 (ABCA1) was one of the most significantly upregulated genes involved in lipid metabolism. Interestingly, a unique axis between microRNA-223 (miR-223) and ABCA1 was identified as a regulator of microglial LDs [78]. Overexpression of miR-223 promoted the expression of ABCA1 in microglia and enhanced the clearance of myelin debris and LDs. This work specifically demonstrated that the accumulation of LLMs in response to SCI is regulated in a lipid metabolism-dependent manner through ABCA1. 

The glucocorticoid receptor (GR), an evolutionarily conserved nuclear steroid receptor, has also recently been implicated in the regulation of LLMs in SCI [79]. Given that glucocorticoids have anti-inflammatory properties and promote recovery after SCI, Madalena and colleagues tested the effect of depleting GR in microglia. Surprisingly, they found that the depletion of GR impaired LD accumulation and myelin phagocytosis, thus dampening the formation of foamy microglia/macrophages marked by Cx3CR1. Together, their findings suggest that GR may be an early regulator of microglial lipid-sensing and phagocytosis. 

*APOE* has long been established as a dominant regulator of lipid metabolism during neurodegeneration. *APOE* is the most abundant lipoprotein in the CNS, primarily expressed by microglia and astrocytes, and plays a key role in cholesterol metabolism [80]. In SCI, *APOE* is a top-upregulated gene in microglia [81]. Additionally, polymorphisms in the human *APOE* gene are strongly associated with chronic pain in SCI [82]. Tansley et al. utilized single-cell RNA sequencing to demonstrate that *APOE*’s regulation of lipid metabolism may directly affect microglial inflammatory functions. Specifically, following the phagocytosis of cholesterol-rich myelin debris, *APOE* promotes the efflux of cholesterol, which accumulates in LDs in microglia. This study showed that *APOE* regulation of LDs in foamy microglia can directly perturb the microglial inflammatory profile. To further assess the role of *APOE* in LLMs in SCI, Yao et al. [81] performed cervical spinal cord hemi-contusion on *APOE-/-* mice. Interestingly, *APOE-/-* mice demonstrated increased microglial LDs and dense lysosomal material in microglia, as visualized by transmission electron microscopy. These mutant mice also had worsened neurological dysfunction, neuroinflammation, and demyelination. These findings agree with the study by Tansley and colleagues, which highlights *APOE* as a key regulator of microglial LDs and inflammation following SCI.

Lipophagy refers to the specific autophagic breakdown of LDs [83]. Given the active role of LLMs, which contain excessive LDs, in SCI progression and resolution, it is likely that regulators of lipophagy can impact the function of LLMs. A study by Wang and colleagues aimed to characterize the role of CD36 in microglial lipophagy [84]. CD36 is a fatty acid translocase that negatively regulates autophagy. Wang et al. [84] established a mouse model of SCI by performing laminectomy at T9-T10 vertebrae, followed by moderate compression injury. They found that microglia with excessive LDs, identified by TEM and Iba1 immunoreactivity, displayed an elevated proinflammatory response, which eventually triggered pyroptosis. These microglia also displayed an increase in the expression of CD36 and breakdown of lipophagy. Conversely, the treatment of mice with sulfo-N-succinimidyl oleate sodium, a CD36 inhibitor, enhanced lipophagy, LD degradation, and SCI recovery. Thus, CD36 regulates lipophagy in microglia following SCI by promoting the uptake of fatty acids. Additionally, Yao et al. [25] utilized a temporal ultrastructural approach to provide thorough characterizations of the molecular changes following SCI. Using this approach, they identified novel epigenetic regulators of LD accumulation in microglia. Specifically, they found that a key feature of chronic SCI was the increase in cholesterol and m6A methylation in LD-accumulating monocytes and microglia.

Taken together, these studies show that LLMs are key regulators of SCI. In addition, the research in the field thus far has shown that there are diverse regulators of the accumulation of LDs in microglia following SCI, including lipoproteins such as *APOE*, microRNAs such as miR-233, and even methylation of lipid genes.

### 6.2. Traumatic Brain Injury

Traumatic brain injury (TBI) occurs when an external mechanical force causes an acquired insult to the brain, leading to temporary or permanent impairment [85]. Zambusi et al. [86] found that, in postmortem cortical brain tissues from patients with TBI, microglial activation was correlated with the accumulation of LDs and TAR DNA-binding protein of 43 kDa (TDP-43+) condensates. The study investigated microglial dynamics in TBI using the zebra fish. Unlike mammals, zebra fish have extensive regenerative potential. Using a stab wound, a zebra fish TBI model was established by destroying the telencephalic hemispheres and 4C4 immunostaining was used to specifically mark microglia. In this study, they also identified a unique injury-induced microglial state characterized by the accumulation of LDs and TDP-43+ condensates, validating the results they found in humans. Interestingly, this state was transient as microglia spontaneously returned to a homeostatic state. This transition was mediated by Granulin as Granulin-deficient microglia were unable to resolve LD accumulation, prolonged microglia activation, and reduced neurogenesis. These findings also directly show that TBI induces the formation of LLMs.

Furthermore, Sridharan et al. [87] provided insights on the role of LLMs in TBI. Following TBI in a mouse model, they observed increased mitochondrial fission accompanied by increased levels of mitochondrial fission 1 protein (Fis1). Pharmacologically preventing Fis1 binding to its partner, dynamin-related protein 1 (Drp1), prevented mitochondrial impairment, microglial activation, and LD formation, which attenuated cognitive impairment and neurodegeneration. These findings suggest that in the context of TBI, LLMs may be pathological and regulated by mitochondrial energy dynamics.

Overall, in TBI, LLM represents a response to injury. This response is mediated by lipid and mitochondrial proteins. If left unmonitored, this response could lead to neurodegeneration. This indicates that, in TBI, the strict regulation of LLM is necessary for injury resolution.

## 7. Lipid-Laden Microglia in Glioblastoma

Glioblastoma (GBM) is the most lethal malignant brain tumor in adults, with an average survival duration of 12 to 14 months [88]. This malignancy lives within a complex tumor microenvironment, where tumor-associated microglia/macrophages (TAMs) play a central role in promoting tumor progression, immune evasion, and therapy resistance [89]. TAMs are heterogeneous populations including both resident microglia and monocyte-derived macrophages recruited from the circulation. These two cell types differ in their developmental origin, spatial localization, and immunological functions. Despite these distinctions, TAMs collectively constitute the most abundant non-neoplastic cell population in the GBM microenvironment, accounting for up to one-third of all tumor-associated cells [90]. However, challenges remain in clearly distinguishing microglia from macrophages in human GBM [89]. As a result, many studies refer to these cells collectively as TAMs and analyze them as a unified population, even though their ontological and functional roles in tumor biology are distinct.

LD accumulation is increasingly recognized as a hallmark of various diseases, including GBM. These LDs play an important role in cancer immunosuppression, drug resistance, aggressiveness, and crosstalk with other cell types in the tumor microenvironment [91]. LD-laden glioma cells augment the secretion of vascular endothelial growth factor (VEGF) and hepatocyte growth factor (HGF), which induces tumor vascularization, glioma-associated microglia/macrophage recruitment, and functional alternations [92]. These recruited microglia/macrophages subsequently undergo functional reprogramming and contribute to the establishment of a highly immunosuppressive microenvironment. This immunosuppressive state not only supports tumor progression but also impairs the efficacy of conventional therapies and correlates with poor patient prognosis [92,93].

Beyond tumor cells themselves, accumulating evidence now reveals that LD enrichment also occurs in immune cells within the GBM microenvironment, particularly in TAMs. Governa et al. [94] characterized a population of macrophages with abundant LDs in human GBM that they called tumor-associated foam cells (TAFs). These cells constitute up to 40% of the total TAM population in GBM. TAFs accumulate LDs through the uptake of extracellular vesicles released by GBM cells, which contributes to their pro-tumorigenic phenotype. Gene set enrichment analysis and immunostaining revealed that the majority of TAFs originate from bone marrow-derived monocytes, while only ~20% are resident microglia.

In parallel, Kloosterman et al. [95] identified a distinct subpopulation of lipid-laden tumor-associated macrophage (TAM), including both brain-resident microglia and infiltrating monocyte-derived macrophages, that are enriched specifically in mesenchymal-like GBM tumors. These cells exhibit an immunosuppressive phenotype marked by elevated CD39 and PD-L1 expression and reduced MHC-II levels. Functionally, they engulf myelin debris from the tumor microenvironment and export lipids to mesenchymal-like GBM cells via an LXR/ABCA1-dependent mechanism, supporting tumor metabolic demands and enhancing proliferation [95,96]. Interestingly, another study reported that the treatment of GBM with a combination of rapamycin and hydroxychloroquine induces substantial LD accumulation in TAMs [97]. Contrary to earlier findings that linked LD accumulation with impaired phagocytic capacity [6], these rapamycin and hydroxychloroquine-induced lipid-laden TAMs displayed increased phagocytosis ability and more proinflammatory phenotype. This discrepancy underscores the context-dependent nature of LD function in immune cells and highlights the need to better understand how lipid metabolism modulates microglial and macrophage behavior in the tumor setting [6,97].

Collectively, LDs play an important role in both tumor cells and TAMs in their tumor microenvironment. LDs are increasingly proposed as novel biomarkers and potential therapeutic targets, particularly in the context of immunometabolic reprogramming and anti-inflammatory strategies for GBM. Current evidence suggests that most lipid-laden TAMs in GBM are derived from bone marrow-derived macrophages rather than resident microglia [95]. However, the line between tumor-associated microglia and tumor-associated macrophages is blurred in many studies due to the lack of a clear distinction between the two populations in brain tumors. The precise contribution of LLMs to brain tumors remains poorly defined and warrants further investigation.

## 8. Lipid-Laden Microglia in Obesity and Diabetes

Metabolic disorders like obesity and diabetes are increasingly recognized as not only peripheral health concerns but also systemic diseases that contribute to neurodegeneration, depression, and anxiety [98]. Clinical studies show that patients with diabetes have a two-fold higher risk of experiencing diabetes-associated cognitive impairment [99]. One study reported that obesity results in the accumulation of senescent glial cells (mostly astrocytes and microglia) in the lateral ventricle, a region critical for adult neurogenesis [100]. These cells establish the phenotype of excessive LD accumulation, termed ‘accumulation of lipids in senescence’ (ALISE). ALISE glial cells were associated with reduced neurogenesis and the prevalence of anxiety. Remarkably, pharmacological or genetic clearance of these senescent cells restored neurogenic potential and alleviated anxiety, which suggests a promising therapeutic avenue for neuropsychiatric disorders [100]. 

As a risk factor for neurodegenerative disorders, type 2 diabetes mellitus (T2DM) causes LD accumulation in microglial cells in the hippocampus but not in neurons in mice [101]. These LDs colocalized with *TREM*1—a microglia-specific inflammatory amplifier. This buildup of *TREM*1 enhanced neuroinflammation by activating the NLRP3 inflammasome, thereby exacerbating cognitive deficits in T2DM [102]. Together, these studies highlight the emerging role of LLM in mediating neuroinflammation, impaired neurogenesis, and cognitive decline in metabolic disorders such as obesity and diabetes.

## 9. Sex Differences and Environmental Effects

Beyond disease-associated triggers, the formation of LLMs is also influenced by biological sex and environmental exposures. Although relatively understudied, emerging evidence suggests that LLM formation exhibits sex-dependent patterns. For instance, there is a study that characterized the sex-dependent response to demyelination in microglia of progranulin-deficient mice [103]. Heterozygous mutations in the *GRN* gene lead to haploinsufficiency of progranulin, which is one of the most common genetic causes of frontotemporal lobar degeneration (FTLD). In male mice, progranulin-deficient microglia accumulate LDs and lipofuscin, increase ROS, and decrease mitochondrial respiration, but these mechanisms are not observed in female mice [103]. These sex-dependent alterations are also consistent with the postmortem analysis of FTLD patients with *GRN* mutations, though the underlying molecular mechanisms remain unclear [103].

Lee et al. [104] further reported that in their tauopathy mouse models, circadian nuclear receptor REV-ERBα depletion in microglia reduced tau uptake, upregulated inflammatory signaling, impaired lipid metabolism, and caused LD accumulation in microglia. This was observed in male mice but not in female mice [104]. Similarly, another study showed that Atg5-mediated autophagy prevents excessive LD accumulation in microglia during AD-related stress, with a notable sex difference [105]. In this case, however, loss of Atg5 in microglia in response to Aβ stimulation enhanced LD accumulation and activation more in female mice than in male mice, suggesting that both autophagic regulation and sex play key roles in modulating microglial lipid homeostasis [105].

Environmental exposures can also lead to LLM formation. Lead pollution, a worldwide public health issue, is associated with neurodegenerative diseases, among others. A recent study showed that lead exposure also disrupts microglial lipid metabolism by damaging lipophagy, which induces LD accumulation, and fatty acid oxidation in microglia [106].

Collectively, these studies suggest that LLM formation is not only the consequence of disease triggers but also affected by intrinsic and extrinsic factors like sex and environmental toxins. These variables should also be considered in future studies in this field.

## 10. Conclusions

In conclusion, as the resident immune cells of CNS, microglia can undergo profound metabolic and functional reprogramming under stress, resulting in the accumulation of intracellular LDs and the formation of LLM. LLM represent a dynamic and multifaceted cellular phenotype that arises across a wide spectrum of physiological and pathological contexts—from aging and neurodegeneration to brain tumors, metabolic dysfunction, injury, sex difference, and environmental exposures. As an emerging hallmark of neuron inflammation, unresolved LLMs are frequently associated with dysfunctional phagocytosis, heightened proinflammatory signaling, and disease progression in various neurodegenerative diseases.

However, emerging evidence reveals that LD accumulation in microglia is a double-edged sword. In some acute stress conditions, such as demyelination or stroke, LLMs may play a transiently protective role, with LDs supporting neuroprotection by activating anti-inflammatory states, promoting repair mechanisms, and clearing debris. Modulating LD dynamics in microglia may therefore offer novel therapeutic strategies. Overall, current evidence suggests that the role of LD formation is dynamic and context-dependent: while moderate LD accumulation at specific stages can be protective, impaired clearance leading to chronic, excessive LD buildup can drive microglial dysfunction.

Additionally, advancements in both label-dependent and label-free imaging technologies have greatly enhanced our ability to track and study LLMs, offering new insight into their dynamics, distribution, and cross-talk with other cells in the CNS. Together, these findings are reshaping our understanding of LLMs and inspiring the development of novel therapeutic strategies.

## 11. Future Directions

Although significant progress has been made in identifying their triggers, signaling pathways, and characteristics of LLMs by various tools, the field still lacks a unified classification framework based on nomenclature, lipid composition, origin, and functional state. Terms such as ‘foamy microglia’, “lipid droplet-accumulating microglia” (LDAMs), and ‘lipid-laden microglia’ are often used interchangeably across different disease contexts, even though these microglial subtypes frequently share similar functional impairments. A standardized and systematic nomenclature is essential for advancing the field. Moreover, the influence of sex and other extrinsic factors on LLMs remains underexplored, and more systematic investigation is needed to achieve a comprehensive understanding.

The lipid composition of LDs in LLMs may vary depending on the inducing condition. A comprehensive classification of LLM subtypes based on lipid content remains lacking, limiting our understanding of their functional heterogeneity. Another persistent challenge in the field is distinguishing LLMs and lipid-laden macrophages. Although both contribute to neurodegeneration, they differ in origin: microglia are CNS-resident immune cells derived from yolk sac progenitors, whereas macrophages are monocyte-derived and infiltrate the CNS during disease. This distinction becomes particularly blurred in conditions like GBM, where both populations coexist within the tumor microenvironment. Furthermore, many studies have focused predominantly on lipid-laden macrophages, especially in diseases like MS. Future studies should therefore more clearly differentiate between macrophages and microglia to ensure accurate functional attribuation.

Surprisingly, LD accumulation in microglia remains poorly characterized in certain neurodegenerative diseases, such as Parkinson’s disease. While lipid dysregulation in neurons has received considerable attention, it is increasingly recognized that neuronal lipid imbalance can indirectly promote LD accumulation in microglia. Future studies that focus on characterizing LDs in microglia, alongside neurons, may offer deeper insights into the mechanisms underlying neurodegeneration.

Moving forward, a deeper understanding of LLM biology could enable the development of targeted strategies to modulate microglial lipid metabolism in a context-specific manner—ultimately opening new therapeutic avenues for a broad range of disorders.

## Figures and Tables

**Table 1 cells-14-01281-t001:** Advantages and disadvantages of emerging tools for detecting and analyzing lipid droplets in microglia.

Tools	Principles	Advantages	Disadvantages
**Label-dependent approaches (invasive)**			
Sudan Black B; ORO ^1^	Lipophilic diazo dyes	Classical stains for visualizing neutral lipids; inexpensive	Only for fixed cells; requires freshly prepared and sensitive to preparation conditions
Dil-oxLDL	Lipophilic, non-toxic fluorescent dye DiI attached to oxidized LDL to trace LDL uptake	Useful for studying the uptake and trafficking of LDL in live cells	Not a direct LDs marker; needs to be combined with other LDs dye for study
BODIPY; Nile Red	Lipophilic fluorescent molecules	Both have high specificity and photostability for LDs; can be applied to both live and fixed cells; Nile Red is a polar-sensitive fluorescent molecule	May have non-specific binding to other lipid-rich membranes like mitochondrial and nuclear membranes
TEM ^2^; SEM ^3^	Electron beam imaging	Better preservation of cellular structures	Cannot be used for live and dynamic imaging; requires cell-fixing
**Label-free approaches (non-invasive)**			
Synchrotron-based microFTIR spectroscopy	Lipid-characteristic infrared absorption spectra	Monitors lipid content and efflux without labeling	Complex and expensive setup for detecting LDs
CARS microscopy ^4^; SRS Microscopy ^5^	Measures molecular specific vibrational frequency	Label-free, real-time, live-cell imaging; high spatial resolution	CARS can achieve higher spatial resolution than SRS; however, SRS provides superior quantification and eliminates the non-resonant background that can interfere with signal specificity in CARS.
ODT ^6^	Based on the RI differences between LDs and cytoplasm	Label-free, live-cell, 3D imaging; can track LD volume, number, and distribution	Low dynamic range
ODT + RI2FL Deep Learning Model	Enables label-free tracking of LD dynamics over time with high temporal resolution	Predicts LD dynamics in live cells without photobleaching; long-term 4D tracking	AI training dataset required

^1^ Oil Red O (ORO). ^2^ Transmission electron microscopy (TEM). ^3^ Scanning electron microscopy (SEM). ^4^ Coherent anti-Stokes Raman-scattering microscopy (CARS microscopy). ^5^ Stimulated Raman-scattering microscopy (SRS microscopy). ^6^ Optical diffraction tomography (ODT).

## Data Availability

No new data were created or analyzed in this study.

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
