# Peer review of "Lipid-Laden Microglia: Characterization and Roles in Diseases"

_cells, 2025, doi:10.3390/cells14161281_

Round 1
Reviewer 1 Report
Comments and Suggestions for Authors
This review addresses a topic of growing interest: the role of lipid-laden microglia (LLMs) in neurological disease. The manuscript does a good job summarizing what is currently known about microglial lipid droplet accumulation, especially in the context of Alzheimer’s disease. However, there are some major conceptual issues that need to be addressed before the paper can be considered suitable for publication.
Discussion of LLMs is presented in isolation, without placing them in the broader context of microglial phenotypes is worrisome. A wide body of literature has characterized microglial diversity, ranging from convenient M1/M2 polarization to more recent categories such as disease-associated microglia (DAM), homeostatic, and neuroprotective subtypes. The manuscript does not acknowledge or incorporate this background, which is essential to understanding where LLMs fit within microglial biology. Without this context, readers are left with the impression that LLMs are a unique or disconnected phenomenon, which oversimplifies a complex and dynamic landscape.
Many of the claims about the involvement of LLMs in diseases beyond Alzheimer’s are speculative. Checking refs it is clear that the AD data are relatively well-supported, while the extension to other disorders (e.g., Parkinson’s, multiple sclerosis, ALS) is often made without citing sufficient primary literature. In several cases, no disease-specific evidence is provided. These claims should either be better substantiated with appropriate references or clearly framed as hypotheses that remain to be tested.
The terminology around LLMs and their distinction from infiltrating macrophages is inconsistently applied. If lipid droplets are used as a defining criterion, then it must be clarified whether these are microglial in origin, perivascular macrophages, or monocyte-derived cells. This ambiguity needs to be resolved to ensure that readers understand what cell populations are being discussed.
Additionally, the review lacks any meaningful discussion of neuroprotective microglial responses. This is a significant omission, especially considering the therapeutic implications that are discussed later in the manuscript. If LLMs are to be considered potential therapeutic targets, then both their damaging and protective roles—under different conditions—should be addressed.
In parallel to addressing these issues the review must :
- Carefully qualify any claims that extend beyond Alzheimer’s disease unless directly supported by evidence.
- Clarify the terminology used to distinguish microglia from other lipid-laden immune cells.
- Acknowledge the significant knowledge gaps that remain, particularly with respect to causality and disease relevance.
Reviewer 2 Report
Comments and Suggestions for Authors
The present review was interesting and clearly presented. However, there were some concerns.
The explanations of Table 1 and Figure 1, not figure legends, were not written in text.
In section 4. Lipid-laden microglia in CNS neurodegeneration, why did the authors pick up these three diseases, Alzheimer's disease, tauopathies, and demyelinating diseases? Some explanations should be added before "Alzheimer's disease".
Also in section 5, the reviewer made same suggestion. Why did the authors describe about spinal cord injury and traumatic brain injury?
In Figure 1, did the order of diseases in the left square have some means? If not, the order should be matched to the text.
"Conclusion" should be separated as a section.
Round 2
Reviewer 1 Report
Comments and Suggestions for Authors
No new specific comments.